# Low-hallucination Synthetic Captions Via Visual CheckList Based Reinforcement Learning for Vision-Language Model Pre-training

## Abstract

Current pre-training of Vision-Language Models (VLMs) relies on large-scale, high-quality alt-text datasets. However, alt-text data is typically short yet noisy, with this issue being more pronounced for non-English languages. To address this limitation, this paper proposes a recaptioning model that rewrites original alt-text data into versions with rich details while maintaining low hallucination rates. The key to mitigating hallucinations lies in a reinforcement learning approach that leverages preference data produced via visual checklists. Leveraging this recaptioning model, we construct X-Recap - a dataset comprising 1 billion synthetic image-caption pairs with low hallucinations. We empirically demonstrate that a VLM pre-trained on X-Recap substantially outperforms its counterpart trained on the original alt-text data, achieving an average performance improvement of approximately 4.6% across 15 vision-language tasks. To facilitate further research in the community, 20% of the X-Recap dataset will be released to the public.

## 1 Introduction

With the enormous power of large language models, remarkable performance gains have recently been achieved in a variety of tasks in natural language processing (NLP), computer vision (CV), and also in cross-modal fields (Brown et al., 2020; Chung et al., 2022; Chowdhery et al., 2022; Touvron et al., 2023; Dosovitskiy et al., 2021; Alayrac et al., 2022; Liu et al., 2023; Achiam et al., 2023; Wang et al., 2024; Hurst et al., 2024). The Large Language Model (LLM) and the Vision Language Model (VLM) are usually equipped with Transformer (Vaswani et al., 2017) as the backbone and then pre-trained with a tremendous amount of unlabeled data. The strong representation ability of the model, the massive amount of data, and the effective means of training make the foundation models powerful for successfully solving the tasks of vision and language.

While the exponential growth of text data has fueled large language models, the pre-training of vision-language models (VLMs) confronts a critical data bottleneck: the scarcity of high-quality, aligned image-text pairs. The prevailing methods for curating these datasets rely on the extraction of web-based alt-text (Sharma et al., 2018; Changpinyo et al., 2021; Ordonez et al., 2011; Schuhmann et al., 2021; 2022; Byeon et al., 2022; Gadre et al., 2023; Liu et al., 2022; Gu et al., 2022). This approach is fraught with inefficiency; its multistage pipeline of cleaning, filtering, and validation not only causes significant data loss but also yields short yet noisy captions that frequently suffer from misalignment with visual content. Critically, as the finite supply of high-quality, naturally occurring pairs is depleted, scaling these extraction methods leads to diminishing marginal returns, with deteriorating data quality hindering VLM training. This bottleneck highlights an urgent need for novel techniques to generate large-scale, high-quality multimodal datasets.

Although recent exploratory investigations into multimodal synthetic data have shown promise for various cross-modal tasks (Li et al., 2023; Betker et al., 2023; Yu et al., 2024; Li et al., 2024b; Awadalla et al., 2024), their efficacy is consistently undermined by a critical flaw: the prevalence of factual hallucinations. As illustrated in Figure 1, captions generated by previous methods are often rife with erroneous details (highlighted in red) or lack sufficient information. To address this limitation, our work introduces a methodology to substantially mitigate hallucinations in synthetic captions. We

subsequently demonstrate that pre-training on our resulting large-scale, low-hallucination dataset yields significant performance improvements for VLMs.

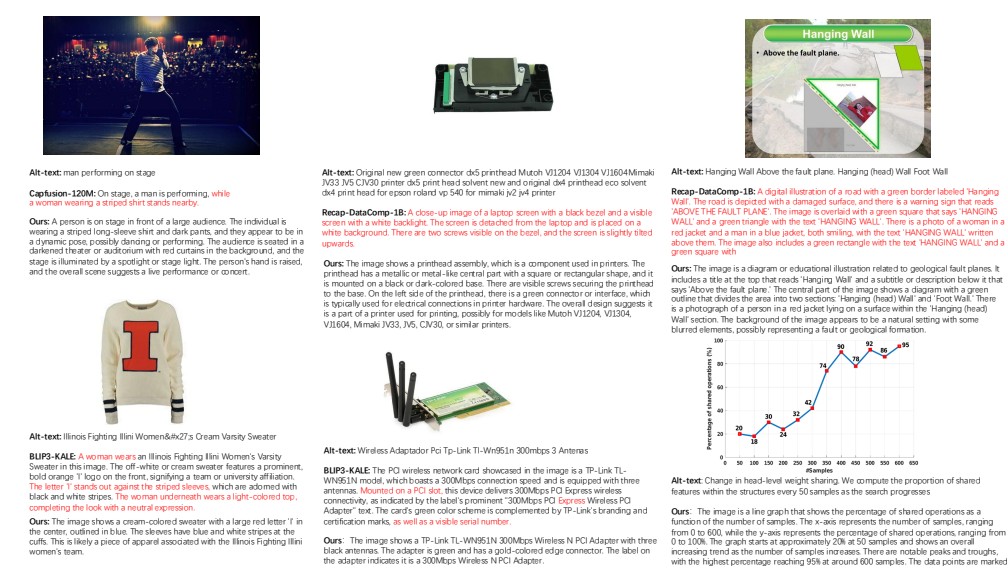

Figure 1: **Examples of the original caption, previous synthetic caption, and our synthetic caption.**

Although VLMs offer an efficient pathway to large-scale caption synthesis, their reliability is critically undermined by persistent hallucinations (Bai et al., 2024). This issue is prevalent in prominent datasets such as Recap-DataComp-1B (Li et al., 2024b) and Blip3-KALE (Awadalla et al., 2024), where, as we show quantitatively in Section 4, significant hallucinatory contents (see Table 1). Such erroneous data corrupt the model's learning process by impeding the acquisition of correct vision-language alignments, making its mitigation a paramount challenge. To address this, we introduce a novel pipeline whose core innovation is the construction of large-scale caption preference data by adjudicating hallucinations against a visual checklist. Constructing large-scale preference captions is a non-trivial challenge that demands a simple, reliable, and scalable production method. To this end, we developed a rigorous model-based framework. Our process first decomposes each caption into fine-grained visual assertions, which a powerful VLM then individually judges for factuality. These assertions act as a visual checklist, enabling the VLM to determine which of two captions contains fewer hallucinations. This "decompose-then-judge" approach is the key to producing high-quality preference data. This preference data then powers an iterative Direct Preference Optimization (DPO) framework. Our approach explicitly aligns the model with preferences for factually grounded descriptions over hallucinatory ones, and we find that it exhibits a distinct scaling law with iterative application. The efficacy of this method is striking: applying our pipeline (detailed in Section 3) to the same dataset used by BLIP3-KALE, we increase the non-hallucination rate from 56.0% to 86.63%. This represents a substantial 30.63 percentage point improvement in data fidelity (metrics in Section 4.1), underscoring the power of the pipeline to create more reliable synthetic captions.

Leveraging our recaption pipeline, we introduce X-Recap, a large-scale synthetic dataset comprising one billion image-text pairs. A primary contribution of this work is the demonstrably high quality of X-Recap, particularly its significantly reduced hallucination rate. We empirically validate the effectiveness of X-Recap as a superior pre-training resource. Systematic comparisons reveal that vision-language models (VLMs) pre-trained on X-Recap consistently outperform those trained on traditional alt-text across various model architectures and data scales. Specifically, our models achieve an average performance gain of at least 4.6% across 15 diverse vision-language tasks under two distinct settings. We attribute this significant improvement to the high information density and factual accuracy of our synthetic data, which enhances the robustness and capabilities of downstream models.

Our contributions are summarized as follows:

1. **A novel pipeline for high-fidelity captions**. We propose a novel pipeline that first constructs a large-scale preference dataset by adjudicating captions against a visual checklist. This data is then

| Dataset | Method | # Visual Details ↑ | # None Hallucinations ↑ | # Hallucination/Visual Details ↓ |
|---------|--------|--------------------|-------------------------|----------------------------------|
| Laion2B | CapsFusion-120M | 2.58 | 72.3% | 0.14 |
| | w Qwen2-VL-7B | 7.00 | 58.01% | 0.094 |
| | w Gemma3-VL-27B | 10.99 | 62.42% | 0.052 |
| | w Qwen2-VL-72B | **7.11** | 69.88% | 0.065 |
| | Our Method (27B) | 6.85 | **79.66%** | **0.040** |
| DataComp | Recap-DataComp-1B | 6.95 | 29.7% | 0.249 |
| | BLIP3-KALE | 6.52 | 56.0% | 0.101 |
| | w Qwen2-VL-7B | 6.35 | 58.02% | 0.109 |
| | w Gemma3-VL-27B | 6.84 | 77.64% | 0.043 |
| | w Qwen2-VL-72B | **8.15** | 74.67% | 0.045 |
| | w Our Method (27B) | 6.83 | **86.63%** | **0.024** |

Table 1: **Hallucination Analysis of Synthetic Caption Datasets**. We compare our method (using Gemma3-VL-27B as the baseline) against existing large-scale synthetic datasets on a random sample of 1,000 pairs from each. We report the Non-Hallucination Rate (% of zero-hallucination captions), Hallucination/Detail Rate (ratio of hallucinatory to total details), and average Visual Details per caption. See Section 4.1 for detailed evaluation methodology.

used within an iterative Direct Preference Optimization (DPO) framework to significantly reduce hallucinations, yielding captions with demonstrably higher factual consistency than prior approaches.

2. **A superior large-scale dataset for VLM pre-training**. We demonstrate that our resulting one billion pair dataset, X-Recap, provides substantial performance gains over existing synthetic data when used for VLM pre-training. Our approach particularly enhances the model's capabilities, providing the community with a valuable, high-fidelity training resource.

## 2 RELATED WORK

**Vision-Language Model.** Vision-Language Models (VLMs) can be broadly classified into two paradigms. The first is the dual-encoder architecture, exemplified by models such as CLIP (Radford et al., 2021) and SigLIP (Zhai et al., 2023b), which learn aligned cross-modal representations through contrastive training and excel at retrieval and zero-shot classification. The second, more recent paradigm integrates a vision encoder with a Large Language Model (LLM), a direction pioneered by Flamingo (Alayrac et al., 2022) and popularized by LLaVA (Liu et al., 2023). Driven by the remarkable success of LLMs (Brown et al., 2020; OpenAI, 2023; Grattafiori et al., 2024; Guo et al., 2025), this architecture has become the focus of extensive research and refinement (Wang et al., 2024; Wu et al., 2024; McKinzie et al., 2024; Chen et al., 2024b; Liu et al., 2024). Despite their architectural differences and the advanced capabilities of modern LLM-based VLMs, both paradigms share a fundamental dependency: a reliance on large-scale, high-quality image-text pairs for effective training. This shared requirement has created a critical bottleneck. The supply of naturally occurring, high-quality image-text data is finite and increasingly exhausted (Villalobos et al., 2022), making it difficult to keep pace with the ever-growing scale of state-of-the-art models. To break this impasse and enable continued progress, synthetic data has emerged as a promising solution.

**Synthetic data.** Building on its success in enhancing large language models (Zhang et al., 2024; Shao et al., 2024; Zhu et al., 2024; Yang et al., 2024), synthetic data generation is increasingly being explored within the multimodal domain. Some methods, such as BLIP-2 (Li et al., 2023), utilize image captioning models to synthesize short captions as substitutes for alt-text, with CapsFusion (Yu et al., 2024) leveraging LLMs for refinement. However, captions generated by these approaches often remain simplistic, providing minimal added information value beyond the original alt-text. Recap-DataComp-1B (Li et al., 2024b) and BLIP3-KALE (Awadalla et al., 2024) attempt to transform alt-text using existing vision language models, but face the significant challenge of hallucination inherent in current VLMs. Other works focusing on dense captioning, including Allava (Chen et al., 2024a) and Dense-Fusion-1M (Li et al., 2024c), generate localized descriptions, but the resulting datasets can also suffer from notable hallucinations. These limitations in existing synthetic VLM data highlight the need for more sophisticated generation techniques. In summary, generating high-quality multimodal synthetic data presents challenges related to controlling hallucination. Although VLM hallucination is a studied topic (Bai et al., 2024), a comprehensive synthetic visual captioning method capable of simultaneously producing diverse and rich content with minimal hallucination is still needed.

Our work introduces a novel approach for generating high-fidelity synthetic captions, culminating in a large-scale dataset with a demonstrably low hallucination rate. We provide comprehensive empirical evidence that this dataset serves as a highly effective resource for large-scale VLM pre-training. This contribution offers a viable and scalable alternative to traditional data sources, effectively addressing the dual challenges of scarcity of high-quality real-world data and the diminishing returns of web-scraped datasets.

## 3 METHODOLOGY

In this section, we detail our synthetic data generation pipeline. We begin by outlining its two core stages: an initial Supervised Fine-Tuning (SFT) phase, followed by an Iterative Direct Preference Optimization (DPO) framework. As a key part of our DPO description, we will elaborate on the novel method used to construct large-scale preference data. The section concludes with the full training procedures and hyperparameter configurations.

### 3.1 SUPERVISED FINE-TUNING

A significant limitation of previous recaption models (Li et al., 2023; Yu et al., 2024) is their tendency to produce generic descriptions that lack specificity. For example, a caption might read "a man performing on stage" (Figure 1), which, due to its limited informational value, offers minimal benefit in training powerful vision-language models. In contrast, our work places a significant emphasis on integrating rich, factual information. To achieve this, we leverage GPT-4o to construct a dataset of captions with high information density, which serves as the training data for our Supervised Fine-Tuning (SFT) stage. This approach compels our model to generate descriptions with a much higher degree of specificity and detail.

Please describe the content of the image in detail with one paragraph, let's proceed step by step:
Step 1: Please describe the content of the image accurately and specifically in one paragraph, avoiding subjective comments (such as "gives a sense of tranquility"). **Entities/knowledge** directly related to the image content (**such as "Eiffel Tower" instead of "a metal tower"** ) should be included in the description. Ensure semantic correctness and fluency of the sentences, and the length of the returned description text should be more than 50 words.
Step 2: Please refer to the image title to improve the description from Step 1. **If the image title contains specific place names, personal names, IP names, etc., and these can be inferred from the image, please add them to the description.** If these details cannot be inferred from the image, do not add them. Do not mention the source of the information in the improved description, and do not include phrases like "image title".
Step 3: Explain the changes made in the description from Step 2.
Please return the result in the following JSON format:
{
  "caption": "the caption",
  "improved_caption": "the improved caption",
  "explanation": "the reason to make the change"
}
Image Title: **a man fishing at sunset on a lake with his guide**

Figure 2: **Prompt Design for Generating informative Captions with GPT-4o**. Our prompt is structured as a three-step procedure designed to elicit informative captions. The instructions highlighted in red are specifically engineered to guide the model in injecting relevant world knowledge.

By constructing SFT data with captions rich in descriptive details, we enable the model, through fine-tuning, to learn to produce detailed and specific descriptions. Therefore, we designed a detailed prompt and used GPT-4o to generate the initial training data. While GPT-4o understands the prompt requirements, the output data still contain a significant number of hallucinations. To address this, we conducted a manual review of the generated captions. By integrating these data processing steps, we constructed the knowledge-enhanced supervised fine-tuning data suitable for training models capable of generating rich image captions, such as the "arctolamia gestro" illustrated in Figure 1.

### 3.2 ITERATIVE DPO

Reinforcement Learning from Human Feedback (RLHF) has proven to be effective in mitigating hallucinations in large language models (LLM) (Zhang et al., 2023; Huang et al., 2025). As a specific and efficient RLHF paradigm, Direct Preference Optimization (DPO) (Rafailov et al., 2023) offers a compelling method for fine-tuning models on preference data. The DPO loss function, shown in Equations 1 and 2, is designed to optimize for this preference directly. As noted in Rafailov et al. (2023), the gradient of the loss function, $\mathcal{L}_{\text{DPO}}$, systematically increases the likelihood of preferred

completions ($y_w$) while decreasing that of dispreferred completions ($y_l$). This mechanism provides a natural and logical framework for combating hallucinations in image captioning. Intuitively, by defining hallucination-free captions as the preferred data ($y_w$) and captions containing hallucinations as the dispreferred data ($y_l$), the DPO process directly trains the model to favor factually grounded descriptions. Maximizing the probability of accurate samples while minimizing the likelihood of hallucinatory ones steers our recaption model toward generating captions with the highest possible fidelity.

$$\mathcal{L}_{\text{DPO}} = -\mathbb{E}_{(x,y_w,y_l)\sim\mathcal{D}}\left[\log\sigma\left(\beta\log\frac{\pi_\theta(y_w|x)}{\pi_{ref}(y_w|x)} - \beta\log\frac{\pi_\theta(y_l|x)}{\pi_{\text{ref}}(y_l|x)}\right)\right] \tag{1}$$

$$\nabla_\theta\mathcal{L}_{\text{DPO}}(\pi_\theta;\pi_{\text{ref}}) =$$

$$-\beta\mathbb{E}_{(x,y_w,y_l)\sim\mathcal{D}}\left[\underbrace{\sigma(\hat{r}_\theta(x,y_l)-\hat{r}_\theta(x,y_w))}_{\text{higher weight when reward estimate is wrong}}\left[\underbrace{\nabla_\theta\log\pi(y_w\mid x)}_{\text{increase likelihood of }y_w} - \underbrace{\nabla_\theta\log\pi(y_l\mid x)}_{\text{decrease likelihood of }y_l}\right]\right],$$
$$\tag{2}$$

where $\hat{r}_\theta(x,y) = \beta\log\frac{\pi_\theta(y|x)}{\pi_{\text{ref}}(y|x)}$ is the reward implicitly defined by the language model $\pi_\theta$ and the reference model $\pi_{\text{ref}}$.

We observe that standard DPO fails to yield continuous performance improvements in our caption synthesis task, quickly reaching a performance plateau. We attribute this to the static implicit reward signal generated by the fixed reference model in the DPO objective function (Equation 2), which causes the policy to prematurely converge to a local optimum. To overcome this, our iterative DPO framework periodically updates the reference model with the current best policy whenever performance saturates. The updated model then generates a new, more challenging set of preference data, providing a refreshed reward signal that enables the policy to resume optimization and surpass its previous performance ceiling.

Furthermore, to ensure that the model learns from the semantic content of the captions rather than superficial features, we address a potential training bias related to sequence length. Since longer completions may have a higher probability of containing hallucinations, we apply a length-balancing operation to the preference pairs during DPO training. This ensures that the average sequence lengths of the preferred ($y_w$) and dispreferred ($y_l$) completions are closely matched, preventing the model from learning a spurious correlation between caption length and quality.

### 3.3 LARGE-SCALE PREFERENCE DATA

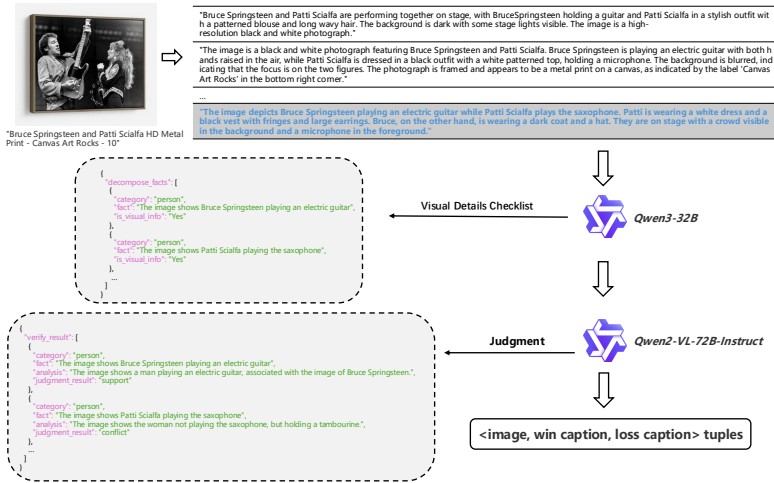

Figure 3: **The illustration of the generating preference captions pipeline.**

The efficacy of DPO is critically dependent on high-quality preference data, yet quantifying caption hallucination is a challenging task. While large-scale human annotation is infeasible due to cost, direct model-based judgments are often unreliable. We therefore introduce a visual checklist methodology for generating superior preference data. Our approach mimics human reasoning: a Large Language Model (LLM, Qwen3-32B) first deconstructs a caption into a checklist of visual details, and a Vision-Language Model (VLM, Qwen2-VL-72B) then verifies the factuality of each detail individually. This two-step process simplifies the detection task.

To validate this methodology, we conducted a controlled experiment. We randomly sampled 200 images and used our initial SFT model to generate eight candidate captions for each image. [1] We then created preference data using two different scoring methods: 1) a baseline approach where a VLM directly scores the entire caption for hallucinations, and 2) our proposed method, which uses the visual checklist-based scoring. As demonstrated in Table 2, our method not only substantially outperforms direct VLM-based evaluation using the same base model but also matches or exceeds the performance of stronger, closed-source VLMs, confirming the superiority of our structured approach. Specifically, when using the same VLM for judgment, our visual checklist-based method improves upon the direct scoring baseline by 23.3% in precision and 36.3% in recall. This significant improvement indicates that, compared to direct, holistic scoring, our visual checklist approach can more comprehensively and accurately identify individual points of hallucinatory information within a caption.

| Model Evaluation | Gemini-2.5-Pro VLM score | GPT4o-latest VLM score | Qwen2-VL-72B VLM score | Qwen2-VL-72B Visual Checklist |
|---|---|---|---|---|
| Precision | 82.8% | 70.8% | 68.2% | **91.5%** |
| Recall | 68.7% | 45.8% | 19.3% | **55.6%** |

Table 2: **Evaluation of Caption Preference Data**. In this table, Precision measures the fraction of retrieved correct preference pairs, while Recall measures the fraction of all existing correct preference pairs that were successfully retrieved.

### 3.4 TRAINING PIPELINE

This section details our comprehensive training pipeline. The pipeline consists of two primary stages: an initial SFT phase, followed by several rounds of iterative DPO. For our experiments, we employ three distinct VLMs as base models: Qwen2-VL-7B, Gemma3-27B, and Qwen2-VL-72B. The entire training process is implemented using the LLaMAFactory (Zheng et al., 2024) framework. A comparative analysis of the characteristics and performance of each baseline is presented in Section 4.

**Supervised Fine-Tuning.** We initiated our pipeline by sampling a raw dataset of image-text pairs from several open-source multimodal corpora, including CC3M (Sharma et al., 2018), CC12M (Changpinyo et al., 2021), DataComp (Gadre et al., 2023), and Wikipedia (Srinivasan et al., 2021). Using the prompt detailed in Figure 2, we then employed GPT-4o to generate an initial set of recaptioned data, from which we retained $43,408$ high-quality entries. Subsequently, we performed Low-Rank Adaptation (LoRA) fine-tuning on the base model using this curated data. Fine-tuning was performed for 10 epochs with a global batch size of 128 and a peak learning rate of $1e-5$, which produced our initial SFT model.

**Iterative DPO.** As outlined in Section 3.2, our methodology begins with the initial SFT model. To generate preference data, we first randomly sample $200,000$ image-text pairs and perform inference using this SFT model. For each pair, we generate eight diverse candidate captions using the following sampling hyperparameters: TOPP=1.0, TOPK=20, and TEMPERATURE=1.0. From this set of eight candidates, we then construct a preference pair $(y_w, y_l)$ by selecting the best (chosen) and worst (rejected) outputs determined by a critic model. We employ Qwen2-VL-72B as our critic model, which operates via a two-step process: it first decomposes each caption into fine-grained visual details and subsequently identifies the number of hallucinations by assessing the factuality of each detail. For each set of candidates, the caption with zero hallucinations is selected as the chosen response $(y_w)$, while the one with the highest number of hallucinations is designated as the rejected response $(y_w)$. To mitigate potential length-induced bias in training, we apply a length-balancing sampling strategy to the initial preference data. This curation step ensures a similar length distribution between the chosen $(y_w)$ and rejected $(y_w)$ captions. We then use this dataset to perform DPO

---

[1]Decoding parameters were set to TOPP=1.0, TOPK=20, and TEMPERATURE=1.0.

training on the SFT model, resulting in our first-stage DPO model. This first-stage DPO model is then used to bootstrap subsequent iterations of data generation. We repeat this process—sampling new image-text pairs, generating preference data using the latest DPO model for inference, and applying our critic model and length-balancing filtration for a total of four iterations. This iterative refinement yields a final set of preference pairs, with the total volume varying slightly for each base model: $357k$ for Qwen2-VL-7B, $340k$ for Qwen2-VL-72B, and $569k$ for Gemma3-VL-27B. [2] The specific hyperparameters for each stage are detailed in Table 3.

|  | Stage-1-SFT | Stage-2-DPO |
| --- | --- | --- |
| Resolution # pixels | [3136, 12845056] | [3136, 12845056] |
| Dataset # Samples | 43K | [357K, 340k, 569k] |
| Finetuning Type | LoRA | LoRA |
| Batch Size, Learning Rate, Learning Epoch | $128, 1e-5, 10$ | $64, 5e-6, 1$ |

Table 3: **Training Configurations for the Recaption Model Stages**. This table details the specifications for each stage of our model's training process, including the vision parameter settings, dataset characteristics, model details, and training hyperparameters.

## 4 EXPERIMENTS

This section presents a comprehensive quality assessment of our recaptioned data. We begin by outlining the evaluation criteria, followed by an analysis across two key dimensions: 1) the level of hallucination present in the captions, and 2) the data's empirical impact on the pre-training of large-scale vision-language models.

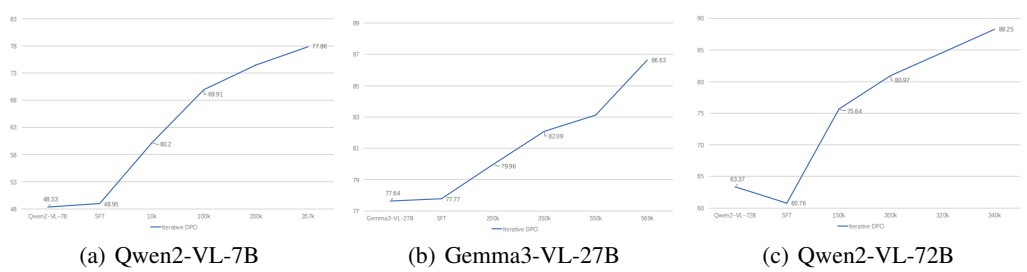

(a) Qwen2-VL-7B     (b) Gemma3-VL-27B     (c) Qwen2-VL-72B

Figure 4: **Performance of Iterative DPO Strategies**. This figure demonstrates the effectiveness of our iterative DPO method in mitigating hallucinations across three different base models. The horizontal axis represents the scale of the training data, while the vertical axis indicates the percentage of hallucination-free captions on the validation set.

### 4.1 CAPTION EVALUATION METRICS

As established in Section 3.3, our visual checklist method is highly effective for generating preference data. We therefore adopt this framework as our primary methodology for the quantitative assessment of hallucination levels. To ensure maximum precision for this critical evaluation task, we leverage a powerful model, GPT-4o, for both the detail decomposition and factuality judgment steps. The reliability of this approach was confirmed through a manual verification study: on a set of 200 randomly sampled cases, two NLP experts concurred with our GPT-4o-based judgments with an accuracy exceeding 95%. We then conduct a comprehensive evaluation to benchmark our method's superiority, validating it on two fronts: 1) against prominent open-source synthetic caption datasets, and 2) against various open-source VLM models.

First, to ensure a fair comparison against other data generation methods, we apply our recaptioning pipeline to the same input image-text pairs used in previous work, including samples from Laion2B (used in CapsFusion (Yu et al., 2024)) and Datacomp (used in Recap-DataComp-1B (Li et al., 2024b) and BLIP3-KALE (Awadalla et al., 2024)). The results presented in Table 1 demonstrate a substantial

---

[2]The dataset for Gemma3-VL-27B is notably larger as it was trained on a multilingual corpus.

reduction in hallucinations. Specifically, our generated captions achieve the following: (1) a 56.9 percentage point increase in the non-hallucination rate compared to Recap-DataComp-1B. (2) an 8.6 and 30.6 percentage point increase in the non-hallucination rate compared to CapsFusion and BLIP3-KALE, respectively.

Second, when compared with other leading open source VLM models on a fixed dataset, our approach again yields a significant decrease in hallucination rates. In both evaluation scenarios, our method consistently produces higher-fidelity captions, proving its superior ability to generate factually grounded training data. Finally, we evaluated the effectiveness of our iterative DPO method for hallucination mitigation. As illustrated in Figure 4, all base models exhibit a significant and consistent increase in the proportion of hallucination-free captions as the DPO dataset is iteratively expanded.

| Dataset | Scientific QA | | | | OCR QA | | | | | Benchmark QA | | | | Hallucination Tasks | | Avg. |
|---|---|---|---|---|---|---|---|---|---|---|---|---|---|---|---|---|
| | MathVista testmini | MMMU val | AI2D test | ScienceQA test | OCRBench test | DocVQA val | TextVQA val | InfographicsVQA val | ChartQA test (aug./hum.) | MMBench en-dev | MMStar test | RealworldQA test | MMVet test | HallusionBench test | None-Hallu. Rate | |
| Alt-text | 50.10 | 45.0 | 73.18 | 85.57 | 59.8 | 77.34 | 68.47 | 43.99 | 62.08 | 72.42 | 47.89 | 61.57 | 42.09 | 33.80 | 39.27 | 57.50 |
| Recap-DataComp-1B | 53.70 | 44.44 | 74.65 | 86.12 | 62.1 | 79.11 | 66.39 | 45.38 | 64.28 | 65.89 | 50.44 | 61.96 | 39.34 | 32.77 | 43.62 | 58.01 |
| Qwen2-VL-7B | 52 | **45.22** | 77.19 | 88.60 | **66.7** | **83.13** | 70.77 | **51.92** | 66.84 | **74.31** | 52.71 | **65.49** | **47.29** | 31.98 | 66.20 | 62.69 |
| X-Recap | **56.20** | 44.00 | **77.91** | **89.74** | 65.9 | 83.06 | **71.37** | 50.32 | **68.96** | 72.34 | **54.66** | **65.49** | 43.17 | **36.38** | **71.86** | **63.42** |

Table 4: **Experimental results on 15 vision-language tasks.** Avg. denotes the average score across all evaluated vision-language tasks. The Non-Hallucination Rate is a hallucination assessment metric defined in Section 4.1. For evaluation, we randomly sampled 1000 instances from the DataComp dataset to quantify the non-hallucination rate. In the table, bold text indicates the optimal performance.

## 4.2 Effectiveness of Recaption Data

Our primary validation assesses the effectiveness of X-Recap for vision-language pre-training at two distinct scales. We first conduct experiments in a small-scale, open-source "toy setting" to ensure rapid community reproducibility. We then perform a comprehensive validation on a larger one-billion-pair dataset to verify that the benefits of our high-quality captions are sustained at scale. Furthermore, we demonstrate the broader extensibility of our data through exploratory experiments on a cross-modal text-to-image generation task.

**Vision-Language Model Pre-Training.**

We first validate the effectiveness of our X-Recap dataset for pre-training Vision-Language Models (VLMs). The objective is to demonstrate that, compared to existing alt-text and other recaption datasets, X-Recap facilitates more effective VLM training, leading to superior performance on both general multimodal benchmarks and our internal knowledge-intensive test sets.

**Experimental Setup**. To ensure a fair comparison, we standardized our evaluation across two distinct experimental settings designed to isolate the impact of different data sources and scales.

1. Open-Source Comparative Setting. In this setting, we aim to directly compare the efficacy of different caption datasets. We adopted the LLaVA (Liu et al., 2023) architecture, utilizing SigLIP (Zhai et al., 2023a) as the visual encoder and Hunyuan-7B (Hunyuan, 2025) as the base large language model. For each dataset under comparison (i.e., various alt-text and our recaption datasets), we sampled a consistent volume of 20M data points for pre-training. Following this, a light, unified supervised fine-tuning (SFT) (Wang et al., 2023; Li et al., 2024a) was applied to all models. It is important to note that our goal here is to isolate the impact of the caption data itself. Therefore, these experiments were conducted at a fixed scale, using only pure caption data without annealing techniques, advanced fine-tuning, or reinforcement learning. As such, the results are intended for comparative analysis and not for direct comparison with state-of-the-art leaderboards.

2. Large-Scale Scaling Analysis. To investigate the scaling properties of our recaptioned data, we conducted a series of large-scale experiments, comparing its effectiveness against traditional alt-text. For this analysis, we used the LLaVA architecture with a smaller Hunyuan-2B language model to balance computational resources and efficiency. The pre-training data for each experimental run was a blend: 70% was either our synthetic captions or the alt-text baseline, mixed with a fixed 30% proportion of a diverse in-house corpus (primarily OCR, STEM, and text-only data). We performed experiments at four distinct scales, corresponding to total token counts of $20B$, $50B$, $200B$, and $1T$. Following each pre-training run, a common 6B-token in-house dataset was used for a unified SFT stage to consistently align and elicit the model's final capabilities.

**Results**. 1. Open-Source Comparative Setting. As presented in Table 4, the Vision-Language Model (VLM) pre-trained on X-Recap significantly outperforms all baselines trained on existing alt-text and synthetic caption datasets across 15 multimodal tasks. This superiority is particularly pronounced on hallucination-related benchmarks, where our dataset provides a substantial performance advantage. For context within the table, "Qwen2-VL-7B" denotes a baseline trained on $20M$ data generated by an unoptimized model with a higher hallucination rate. Furthermore, the benefits of X-Recap extend beyond final performance metrics; as detailed in Appendix A, models pre-trained with our data also exhibit markedly improved convergence properties during the subsequent SFT phase.

2. Large-Scale Scaling Analysis. As presented in Table 5, our recaptioned data demonstrates superior performance compared to the original alt-text data. This advantage is consistent across various data scales and on nearly all evaluation tasks. Crucially, as the data volume increases, X-Recap exhibits a favorable scaling law, indicating that its benefits are sustained and amplified at scale. This superiority is particularly pronounced on hallucination-related benchmarks, a finding that aligns with our results from the Open-Source Comparative Setting and further underscores the significant advantages of low-hallucination synthetic captions. Furthermore, we include the performance of a 2B-scale leading model, Qwen2-VL-2B, for reference. It is important to note that our training methodology was intentionally simplified to isolate the effect of the caption data, without the extensive fine-tuning and optimization typically applied to benchmark models. Despite using less training data and a non-optimized strategy, our model trained on X-Recap achieves better performance than this strong baseline, underscoring the high quality and efficacy of our generated captions.

| Dataset | Data size | Scientific QA | | | | OCR QA | | | | | Benchmark QA | | | | Hallucination Tasks | | Avg. |
| | | MathVista testmini | MMMU val | AI2D test | ScienceQA test | OCRBench test | DocVQA val | TextVQA val | InfographicsVQA val | ChartQA test (aug./hum.) | MMBench en-dev | MMStar test | RealworldQA test | MMVet test | HallusionBench test | None-Hallu. Rate – | |
| --- | --- | --- | --- | --- | --- | --- | --- | --- | --- | --- | --- | --- | --- | --- | --- | --- | --- |
| Qwen2-VL-2B | 1.4T | 43 | 41.1 | 74.7 | 80.4 | 80.9 | 90.1 | 79.7 | 65.5 | 73.5 | 74.9 | 48.0 | 62.9 | 49.5 | 41.7 | 54.33 | 64.02 |
| ALT-Text | 20B | 43.4 | 34.56 | 53.34 | 70.02 | 64.7 | 72.55 | 66.84 | 39.73 | 66.32 | 57.73 | 37.88 | 47.84 | 39.03 | 29.92 | 25.05 | 49.88 |
| X-Recap | 20B | 44.5 | 36.56 | 55.57 | 71.49 | 72.2 | 80.28 | 70.9 | 49.99 | 70.92 | 60.65 | 41.1 | 49.67 | 44.63 | 32.9 | 41.73 | 54.87 |
| ALT-Text | 50B | 41.3 | 34.89 | 55.28 | 71.05 | 71.5 | 81.24 | 72.05 | 49.56 | 70.88 | 62.42 | 40.09 | 53.86 | 42.84 | 35.06 | 36.37 | 56.00 |
| X-Recap | 50B | 46.6 | 36.11 | 59.42 | 74.42 | 73.8 | 82.67 | 73.43 | 52.91 | 72.72 | 62.97 | 42.5 | 55.69 | 43.39 | 33.37 | 49.90 | 57.33 |
| ALT-Text | 200B | 44.6 | 35.33 | 58.1 | 72.88 | 79.1 | 84.33 | 77.14 | 53.66 | 71.6 | 61.77 | 41.3 | 51.37 | 40.41 | 34.63 | 43.29 | 56.63 |
| X-Recap | 200B | 50.7 | 36.22 | 58.42 | 74.91 | 81.4 | 87.08 | 77.43 | 58.48 | 74.56 | 62.97 | 45.58 | 52.94 | 43.48 | 36.37 | 61.72 | 60.36 |
| ALT-Text | 1T | 52 | 36.78 | 61.01 | 74.96 | 82.5 | 87.61 | 78.74 | 57.99 | 76.32 | 62.2 | 45.72 | 52.68 | 48.66 | 40.38 | 48.40 | 60.42 |
| X-Recap | 1T | 54.1 | 37.33 | 68.62 | 76.3 | 85 | 91.36 | 79.57 | 66.91 | 79.2 | 67.01 | 51.74 | 56.47 | 50.0 | 44.39 | 67.63 | 65.04 |

Table 5: Experimental results on 15 vision-language tasks for the Large-Scale Scaling Analysis. Results in bold indicate the superior performance between the two methods under the same data scale setting. An underline denotes the best overall result across all methods, including the comparison with a leading, similarly-sized open-source model, Qwen2-VL-2B.

**Text-to-Image Model Training.** We also validated X-Recap on the text-to-image generation task by fine-tuning the Hunyuan-DiT model (Li et al., 2024d). The fine-tuned model achieved a significantly lower FID and a higher CLIP score than the public baseline (Table 6), confirming that our data enhances the generation of authentic, high-fidelity images. Examples are presented in Appendix B.

| Dataset | FID↓ | | | CLIP Score↑ | | |
| | Row Caption | Our Recaption | Fine-tuning with Our Recaption | Row Caption | Our Recaption | Fine-tuning with Our Recaption |
| --- | --- | --- | --- | --- | --- | --- |
| MSCOCO Test set | 28.79 | 21.09 | **15.46** | 0.307 | 0.305 | **0.313** |

Table 6: **Text-to-Image Evaluation** of the Hunyuan-DiT model on MSCOCO-30K, comparing performance with and without fine-tuning using our recaption data. It should be noted that, due to computational resource limitations, we only sampled $2M$ data points from the X-Recap dataset for LoRA fine-tuning.

## 5 CONCLUSION

This work tackles the critical challenge of hallucination in large-scale synthetic image captioning. Our core finding is that Direct Preference Optimization (DPO) is exceptionally effective at reducing caption hallucinations. We further show that by applying DPO iteratively, we can progressively increase the ratio of hallucination-free captions in our synthetic data, a process that demonstrates a distinct and predictable scaling law. Leveraging this methodology, we construct X-Recap, a new large-scale dataset of high-fidelity, low-hallucination captions. Our extensive experiments demonstrate that pre-training Vision-Language Models (VLMs) on X-Recap yields substantial performance gains over models trained on both raw alt-text and existing synthetic datasets. To foster future research in multimodal learning, we will release the X-Recap dataset to the open-source community. For a detailed discussion of limitations and potential risks, refer to Sections C and D.

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

## A    LOSS OF VISION-LANGUAGE MODEL TRAINING

To address the issue of loss value incomparability arising from diverse pre-training data distributions, we examine the convergence behavior of pre-trained models when fine-tuned on a common supervised dataset (Wang et al., 2023; Li et al., 2024a), as illustrated in Figure 5. Figure 5 shows that models trained with X-Recap data exhibit superior convergence. Quantitatively, these models converge to a lower final loss of $0.4796$ compared to models pre-trained on other datasets.

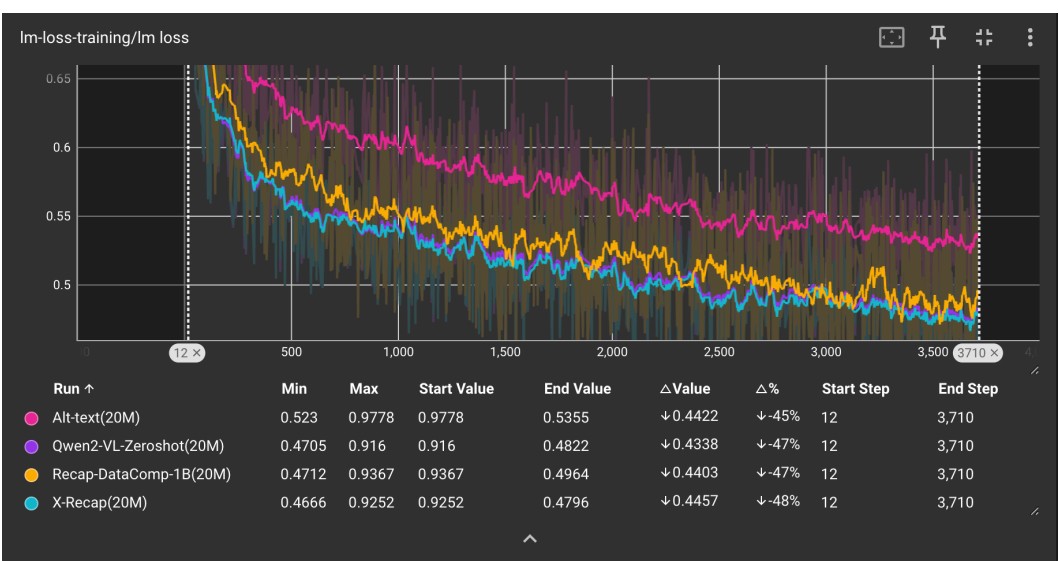

Figure 5: **Loss curve of vision-language model pre-trained with alt-text, Recap-DataComp-1B, Qwen2-VL-Zeroshot or X-Recap.** To ensure a fair comparison, all pre-trained models analyzed here utilized the same amount of data ($20M$) for pre-training. Furthermore, the four convergence curves presented are obtained by fine-tuning these models on an identical supervised dataset.

## B    EXAMPLES OF TEXT-TO-IMAGE GENERATIONS

This section presents examples of text-to-image generation in Figure 6; the corresponding text prompts are provided in Table 7. Observation of these examples reveals that the original Hunyuan-DiT model struggles to accurately render certain concepts (e.g. abacus, Yueqin) and produces less accurate depictions for others (e.g. jellyfish, acerola). However, after fine-tuning with the X-Recap data, the model's understanding of these concepts becomes more precise, resulting in generated images with a more realistic style. These qualitative improvements are consistent with the quantitative results shown in Table 6, such as the observed reduction in FID. This consistency highlights how the comprehensive concept coverage in our data enhances the model's perceptual fidelity, which is further corroborated by the significant performance improvement of the VLM trained with X-Recap on numerous vision-language tasks.

## C    LIMITATIONS

The pre-training of large vision-language models is an inherently resource-intensive endeavor. In our work, for instance, pre-training the 2B-scale VLM on $1B$ tokens alone required $123,776$ A800 GPU hours. Due to these substantial computational demands, a key limitation of this study is that we were unable to explore the full scaling potential of our recaptioned data with even larger VLM architectures. We leave this promising direction for future investigation.

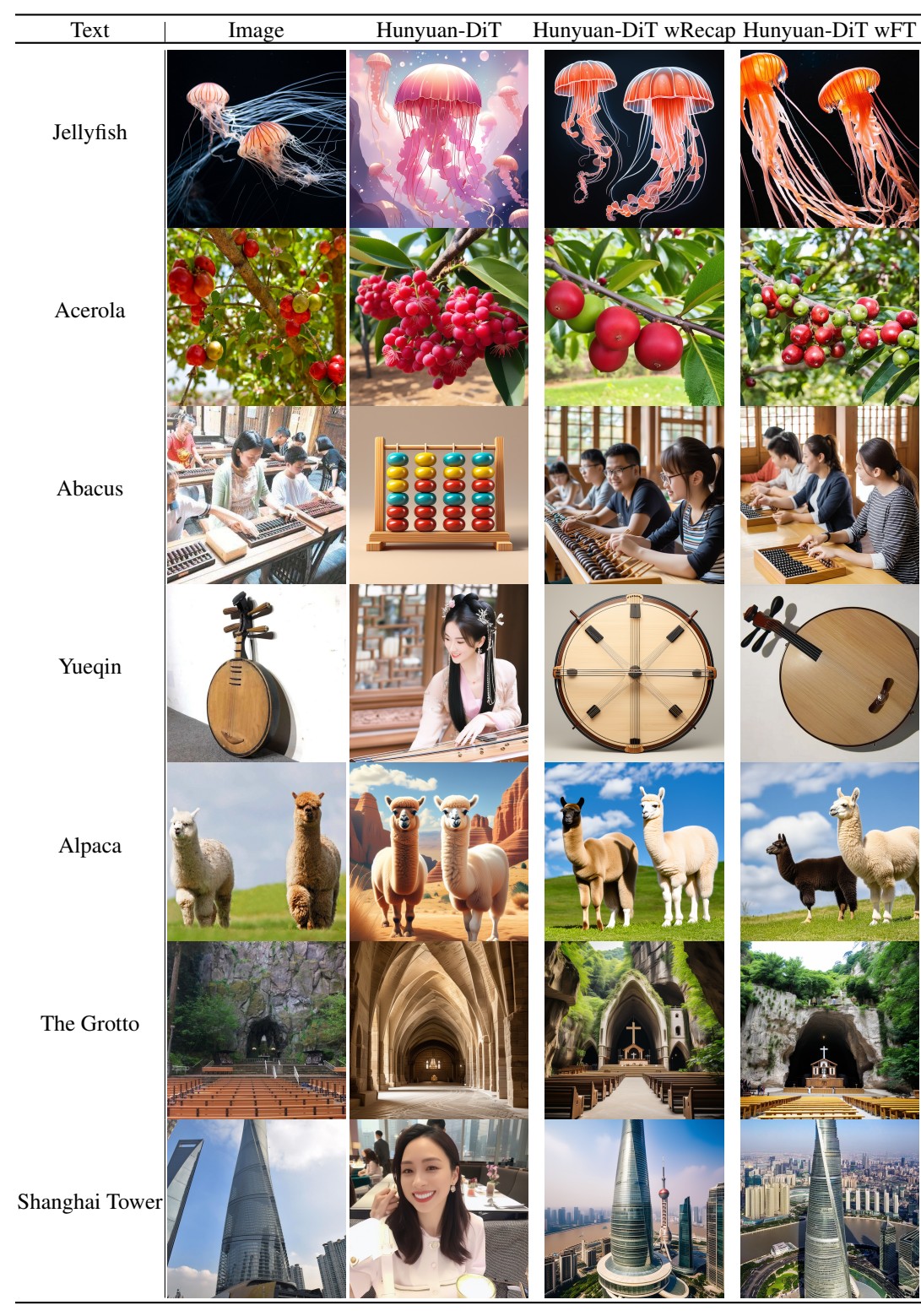

Figure 6: **Demonstration of text-to-image cases.** Here, the original instruction for image generation is denoted as "Text", the original picture as "Image", the generation effect of the original open-source model as "Hunyuan-DiT", the generation effect of the original open-source model with our Synthetic texts as "Hunyuan-DiT wRecap", and the effect of the model fine-tuned with our recaption data as "Hunyuan-DiT wFT".

| Text | Our Recaption |
|------|---------------|
| Jellyfish | The picture shows two jellyfishes. Their bodies are orange or light red with distinct stripes or lines. Their tentacles are long, slender, transparent or semi-transparent. The background is black, highlighting the colors of the jellyfish and the details of their tentacles. The jellyfish are swimming or floating in the water, and their tentacles are spread out in the water, forming a distinct linear effect. |
| Brazilian Acerola | The image shows the branches of a tree laden with red and green fruits. The red fruits look ripe, while the green ones appear unripe. There are also green leaves on the branches. In the background, more green leaves and a few blurred flowers or small buds can be seen. The overall environment seems to be an outdoor natural setting. |
| Abacus | There is a group of people sitting in front of a wooden table in the picture, using abacuses for learning or operation. There are several abacuses on the table. Wooden doors and windows can be seen in the background, indicating that this is an indoor environment, perhaps a classroom or a study place. People are wearing casual or semi-formal clothes, looking relaxed. |
| Yueqin | This is a Yueqin, which has a circular resonator and four strings. There are multiple positions for pressing the strings on the body of the instrument, and there are four tuning pegs on the headstock. Overall, it exhibits the characteristics of a traditional musical instrument. |
| Alpaca | There are two alpacas standing on the grass in the image, with blue sky and white clouds in the background. The alpaca on the left has lighter fur, while the one on the right has darker fur. The grass is green, and the postures of the alpacas suggest that they are either walking or standing. |
| The Grotto | The Grotto is a church located in a natural environment. The facade of the church is a huge rock cave, covered with green vegetation above the cave, and surrounded by dense trees. Inside the church, there is an altar, above which a cross is hung, surrounded by lighting. In front of the church, there is a row of wooden benches for believers to sit and pray. |
| Shanghai Tower | Shanghai Tower is a skyscraper with a spiral glass curtain wall design on its exterior. At the top, there is a circular observation deck, surrounded by other high-rise buildings. The overall architectural style is modern and futuristic. |

Table 7: The original text of the image in Figure 6 and the synthetic caption after being processed by our recaption model.

## D    POTENTIAL RISKS

The original image-text pairs are primarily derived from open-source datasets. While we have implemented substantial measures to filter out undesirable content, potential risks remain. These risks are particularly salient in the field of image generation, exemplified by issues such as the creation of fake portraits for social media (Hill & White, 2020), which is a recognized challenge in this research area.

## E    DECLARATION ON THE USE OF AI TOOLS

During the preparation of this manuscript, the authors utilized a large language model to assist with language editing and refinement. The tool was employed to improve grammar, clarity, and readability. The authors take full responsibility for the final content and all scientific claims presented herein.

