# OpenReview forum: "Low-hallucination Synthetic Captions Via Visual CheckList Based Reinforcement Learning for Vision-Language Model Pre-training"
_ICLR.cc/2026/Conference — ICLR 2026 Conference Withdrawn Submission_

### Official Review · Reviewer_ypcH · 2025-10-27

**Soundness:** 2
**Presentation:** 2
**Contribution:** 2
**Rating:** 2
**Confidence:** 3

**Summary:**

To overcome the short and noisy nature of alt text used for large-scale VLM training, this paper proposes a RL-based recaptioning technique and uses it to construct X-Recap, a dataset of 1B images with synthetic captions. Results suggest that the recaptioning technique shows reduced hallucinations compared to alternative methods, and that training on X-Recap outperforms training on the original alt text or other recaptioning datasets.

**Strengths:**

With the rise of large-scale VLM training, methods for curating high-quality large-scale training data has become increasingly important and errors in training data are a concern.
Overall, the suggested RL method does appear to be effective in reducing hallucinations, and training on the recaptioned data does seem to improve VLM performance.
The release of a large set of recaptioned image-caption pairs is an important contribution to the community.

**Weaknesses:**

Important details of the method are not clearly described, impairing reproducibility. The central “visual checklist” concept, presented as a novel method (L272) and shown in Fig 3 is not clearly defined. The definition of “visual details” (L274) used in the checklist is important, as it is unclear if this only captures object hallucinations, visual attributes, actions, relations between entities, etc. However we are not provided with the prompt used to extract them, or a validation or analysis of the visual details that this captures or misses.

The novelty of the method is limited, as prior works have proposed using VLMs to extract lists of visual details for hallucination benchmarking and mitigation [1–3], reinforcement learning to mitigate hallucinations [2–3], and training on recaptioned data.

There is missing a comparison to existing hallucination mitigation methods and evaluation on existing hallucination benchmarks [4] beyond HallusionBench, while the evaluation used is ad-hoc and potentially circular since it uses a similar methodology to the metric being optimized in the RL training (L367). While training on recaptioned data vs. original alt text is ablated, it is unclear whether training on post-RL recaptioning outperforms training on recaptioned data with the original SFT model, an ablation needed to determine whether the proposed hallucination mitigation method actually affects downstream VLM performance. Most experiments lack human evaluation, leaving it unclear whether automated metrics alone reflect real-world performance.

The formatting can be improved. In particular, the text in Figs 3-4 and Tab 5 is small and hard to read.

[1] Li et al. Evaluating Object Hallucination in Large Vision-Language Models. EMNLP 2023

[2] Ben-Kish et al. Mitigating open-vocabulary caption hallucinations. EMNLP 2024

[3] Yu et al. Rlaif-v: Open-source ai feedback leads to super gpt-4v trustworthiness. CVPR 2025.

[4] Chen et al. A Survey of Multimodal Hallucination Evaluation and Detection.

**Questions:**

Why will you not release the full X-Recap dataset? (L25) Will your code be openly released?

Do the models being trained in Sec 3.4 take (image, text) as input and output rewritten text? Or are they being used as captioning models that only take an image as input. If the latter, is the alt text effectively discarded?

Is iterative DPO with reference model updating a novel method? How does this compare to existing methods [4–5] (in particular, [5] propose occasionally setting \pi_{ref} to \pi_\theta during training).

[4] Wu et al. AlphaDPO: Adaptive Reward Margin for Direct Preference Optimization. ICML 2025

[5] Gorbatovski et al. Learn Your Reference Model for Real Good Alignment. ICLR 2025

---

### Official Review · Reviewer_s1NY · 2025-10-28

**Soundness:** 3
**Presentation:** 2
**Contribution:** 3
**Rating:** 4
**Confidence:** 3

**Summary:**

The authors propose a recaptioning pipeline to generate large-scale, detailed, and low-hallucination synthetic captions, named X-Recap. A key component is the generation of preference data for DPO using a novel visual checklist method, where an LLM decomposes captions into verifiable details, and a powerful VLM judges the factuality of each detail to determine preference. Pre-training VLMs on the resulting X-Recap dataset is shown to outperform models trained on original alt-text and other synthetic datasets like Recap-DataComp across 15 benchmarks.

**Strengths:**

1.The work constructs a large-scale synthetic caption dataset, X-Recap, and demonstrates its superiority for VLM pre-training compared to baseline alt-text data and the existing large-scale synthetic dataset Recap-DataComp.

2.The paper introduces a novel visual checklist method for generating preference data to evaluate caption hallucinations. This method, which involves decomposing captions into details and having a VLM verify each one, is shown to be more effective than having a VLM score the overall caption directly.

**Weaknesses:**

1.The overall methodology appears heavily engineering-focused with limited algorithmic novelty. Within the data generation pipeline, only the visual checklist method for creating preference data seems truly novel. The subsequent model training stages employ standard SFT and DPO techniques, which feels more like a careful implementation rather than a conceptual breakthrough.

2.It's unclear if the pre-training gains from X-Recap stem primarily from the detailed nature of the captions or the hallucination reduction effort via DPO. Since X-Recap was used as pre-training data and not directly for DPO on the final evaluated models, it seems plausible that the benefit might largely come from a basic visual checklist verification filtering egregious hallucinations during data creation, rather than the sophisticated DPO alignment process itself significantly enhancing the data quality for pre-training beyond simply being detailed and mostly correct.

3.The evaluation of hallucination is relatively narrow, relying primarily on the proposed visual checklist based Non-Hallucination Rate and results on HallusionBench. Incorporating established benchmarks specifically focused on object hallucination, such as CHAIR and POPE, would provide a more comprehensive assessment of the dataset's low-hallucination claim.

4.There is a lack of transparency regarding the hallucination judge model used during the large-scale 1B data generation process itself. Was this judge model specifically trained or evaluated for reliable hallucination detection at scale, and what measures were taken to ensure its judgments were consistent and accurate across such a vast dataset?

**Questions:**

Please refer to the Weakness.

---

### Official Review · Reviewer_6rWH · 2025-10-31

**Soundness:** 3
**Presentation:** 2
**Contribution:** 3
**Rating:** 2
**Confidence:** 3

**Summary:**

This paper presents Low-Hallucination Synthetic Captions via Visual Checklist-Based Reinforcement Learning, a pipeline designed to generate large-scale, high-fidelity synthetic image captions with minimal hallucination. The proposed method first constructs preference data using a visual checklist approach, where captions are decomposed into fine-grained visual assertions, and each assertion is individually verified by a Vision-Language Model (VLM). These judgments form the foundation for iterative Direct Preference Optimization (DPO) training that progressively aligns caption generation toward factual accuracy. Using this approach, the authors build X-Recap, a dataset containing one billion image-caption pairs with an 86.63% non-hallucination rate—significantly higher than prior synthetic datasets such as Recap-DataComp-1B or BLIP3-KALE. When used for pre-training multiple VLM architectures (e.g., LLaVA, Hunyuan-VL), X-Recap yields consistent performance gains across 15 multimodal benchmarks and improves convergence in downstream fine-tuning tasks.

**Strengths:**

The paper introduces a visual checklist-based preference construction mechanism, combining LLM caption decomposition with VLM verification to automatically label the preference pairs. This idea is reasonable.

The X-Recap dataset can potentially benefit the community.

**Weaknesses:**

The paper lacks crucial methodological clarity in its core components. For instance, it is unclear how exactly captions are decomposed into atomic “visual assertions” for checklist construction—what linguistic rules, parsing model, or heuristics are used to identify and segment these elements? Similarly, the process by which each visual detail is verified by a Vision-Language Model (VLM) is underspecified: How are verification prompts designed? Are judgments binary or confidence-weighted? What is the output of the judgment? The paper also mentions a length-balancing operation during DPO training but provides no formal description of how this balancing is computed or applied to preference pairs. Without these implementation details, reproducibility and interpretability are significantly weakened.

The use of Supervised Fine-Tuning (SFT) followed by Direct Preference Optimization (DPO) constitutes a well-established paradigm in both language and vision-language alignment. The novelty here rests primarily on the visual checklist-based data curation rather than the optimization pipeline itself. However, the paper does not sufficiently explain what exactly the checklist data curation is. As a result, the writing may need to be improved.

The controlled experiment in Section 4.3 (line 278) and Table 2 (“Evaluation of Caption Preference Data”) is not clearly presented. It remains ambiguous how the experiment was set up, what the “two different scoring methods” actually refer to, and how precision and recall were calculated in your experiment (e.g., the paper says that they generate eight candidate captions for each image, but it is unclear how this relates to Precision and Recall. The source of ground truth labels used for computing these metrics may need to be elaborated). Without explicit procedural details, Table 2 cannot be reliably interpreted or reproduced.


The approach seems to assume that the verifying VLM provides objective and accurate judgments. However, if either the LLM used for decomposition or the VLM used for verification is itself hallucination-prone, systematic biases may propagate into the preference dataset. Furthermore, it is unclear how good the caption decomposition step (LLM hallucination) and the visual verification step (VLM misjudgment) are. An analysis of these components can improve the quality of the paper.

The paper repeatedly claims that the proposed method reduces hallucination, yet it does not explicitly define what type of hallucination it targets. The term “hallucination” can refer to various phenomena—such as object hallucination (nonexistent entities), attribute hallucination (incorrect properties), relational hallucination (wrong spatial or logical relations), ... Without a precise definition and consistent evaluation scope, it is unclear whether the proposed approach mitigates all forms of hallucination or only improves factual grounding at the object level.

**Questions:**

Could the authors describe in detail how a caption is decomposed into a checklist of visual assertions? How do you ensure that non-visual phrases (e.g., relational or emotional terms) are filtered out?

How exactly is each visual assertion verified by the VLM?

The paper mentions a length-balancing operation for preference pairs during DPO training—could you clarify how this balancing is implemented?

How was the controlled experiment for Table 2 set up? Could you explain in more detail the two scoring methods being compared? How are precision and recall defined and computed, and what ground truth annotations were used?

Could the authors explicitly define the type(s) of hallucination addressed (e.g., object, attribute, relational, ...) and clarify how these are operationalized or mitigated in the experiments?

---

### Official Review · Reviewer_oHGi · 2025-11-01

**Soundness:** 3
**Presentation:** 2
**Contribution:** 2
**Rating:** 2
**Confidence:** 3

**Summary:**

The paper tackles the problem that large-scale vision–language models are usually pre-trained on web image-text pairs that are short, noisy, and often describe things not actually in the image, which later shows up as hallucinations at inference time. To fix this, the authors build a scalable recaptioning pipeline: for each image they generate multiple long, detailed captions, decompose each caption into atomic visual facts with an LLM, use a strong VLM to check which facts are really supported by the image (visual checklist), and then form win/lose caption pairs to train the captioner via iterative DPO so it progressively prefers grounded over hallucinatory descriptions. Applying this recipe to the same image sources that prior synthetic datasets used, they construct X-Recap, a 1B-scale dataset whose captions achieve an 86.63% non-hallucination rate, which is higher than earlier recaptioned data, and they show that pretraining VLMs on X-Recap consistently outperforms pretraining on raw alt-text across 15 vision–language benchmarks.

**Strengths:**

* The paper targets a well-recognized bottleneck: VLMs depend on large, high-quality, aligned image–text pairs, but web alt-text is short and often off-image.

**Weaknesses:**

* The proposed large-scale, low-hallucination recaptioned dataset is a useful resource for the community and the empirical results indicate it can improve downstream VL training. However, most components of the pipeline (LLM-based caption expansion, fact decomposition, VLM-based verification, preference learning/DPO) have appeared in prior work individually. The paper’s main advance is therefore in combining these ingredients and scaling them, rather than in introducing a fundamentally new technical idea.

* Decomposing every candidate caption and verifying every atomic fact is clearly more expensive than single-shot scoring, but the paper does not provide a concrete cost/latency analysis or ablation on how much of the pipeline is actually needed at scale.

**Questions:**

See weaknesses above.

---

### Note · Authors · 2025-11-25

I have read and agree with the venue's withdrawal policy on behalf of myself and my co-authors.